# A General Protocol for Electrospun Non-Woven Fabrics of Dialdehyde Cellulose and Poly(Vinyl Alcohol)

**DOI:** 10.3390/nano10040671

**Published:** 2020-04-02

**Authors:** Slavica Hell, Kousaku Ohkawa, Hassan Amer, Antje Potthast, Thomas Rosenau

**Affiliations:** 1Division of Chemistry of Renewable Resources, Department of Chemistry, University of Natural Resources and Life Sciences, Tulln, Konrad-Lorenz Straße 24, 3430 Tulln, Austria; slavica.hell@gmail.com (S.H.); hassan.amer@boku.ac.at (H.A.); antje.potthast@boku.ac.at (A.P.); 2Division of Biological and Medical Fibers, Institute for Fiber Engineering, Interdisciplinary Cluster for Cutting Edge Research, Shinshu University, Ueda 386-8567, Nagano Prefecture, Japan; 3Division of Synthetic Polymers, Institute of High Polymer Research, Faculty of Textile Science and Technology, Shinshu University, Ueda 386-8567, Nagano Prefecture, Japan; 4Department of Natural and Microbial Products Chemistry, National Research Centre, 33 AlBohous St., Dokki, Giza 12622, Egypt; 5Johan Gadolin Process Chemistry Centre, Åbo Akademi University, Porthansgatan 3, FI-20500 Åbo/Turku, Finland

**Keywords:** electrospinning, dialdehyde cellulose, poly(vinyl alcohol), non-woven fabrics

## Abstract

In the past two decades, research on electrospinning has boomed due to its advantages of simple process, small fiber diameter, and special physical and chemical properties. The electrospun fibers are collected in a non-woven state in most cases (electrospun non-woven fabrics, ESNWs), which renders the electrospinning method an optimum approach for non-woven fabric manufacturing on the nano-scale. The present study establishes a convenient preparation procedure for converting water-soluble dialdehyde cellulose (DAC) into DAC-based electrospun non-woven fabrics (ESNWs) reinforced with poly(vinyl alcohol) (PVA). The aldehyde content, which was quantified by colorimetry using Schiff’s reagent, was 11.1 mmol per gram of DAC, which corresponds to a conversion yield of ca. 90%. DAC is fully water-soluble at room temperature between 10 and 30 wt%, and aqueous solutions turn into hydrogels within 24 h. To overcome gelation, NaHSO_3_, which forms bisulfite adducts with aldehyde functions, was added to the DAC and its concentration was optimized at 1 wt%. The electrospun (ES) dope containing 5 wt% DAC, 5 wt% PVA, and 1 wt% NaHSO_3_ in an aqueous solution was successfully transformed into ESNW, with an average fiber diameter of 345 ± 43 nm. Post-spinning treatment with excess hexamethylene diisocyanate was performed to insolubilize the ESNW materials. The occurrence of this chemical conversion was confirmed by energy-dispersive X-ray elemental analysis and vibrational spectra. The cross-linked DAC/PVA ESNW retained its thin fiber network upon soaking in distilled water, increasing the average fiber diameter to 424 ± 95 nm. This suggests that DAC/PVA-ESNWs will be applicable for incorporation or immobilization of biologically active substances.

## 1. Introduction

Electrospun non-woven fabrics (ESNWs) of polysaccharides and their derivatives are of great interest for a wide variety of applications including medical, separation, and filtration technologies [1,2,3]. Two different manufacturing strategies—direct electrospinning [4,5] and post-spinning treatment of ESNWs of precursor polymers [6,7]—are always applied, albeit to different extents, depending on the material to be fabricated.

In the case of polysaccharide-based ESNWs, for instance, we have reported direct electrospinning methods for chitosan [4,5], cellulose [8], and their composites [9] that use trifluoroacetic acid as a spinning dope solvent. Earlier attempts frequently utilized polysaccharide derivatives such as cellulose acetates for electrospinning since the precursors are soluble in conventional organic solvents such as *N,N*-dimethylacetamide and acetone mixtures, which are readily adaptable to conventional electrospinning processes. After preparing the ESNWs from the polysaccharide derivative precursor, a post-spinning alkaline treatment regenerates the polysaccharide ESNWs by reconverting the derivative into the parent polysaccharide. Regenerated cellulose ESNWs are composed of thin fibers, have diameters in the nano- or submicrometer ranges, and are nano-scale materials obviously unlike functionalized cellulose nano-whiskers or nano-crystals [10,11].

Regenerated cellulose ESNWs are chemically modified by partial oxidation with sodium periodate to produce aldehyde moieties on cellulose ESNW matrices, which serve as chemical anchors for the immobilization of functional ligand substances [12]. The oxidation of cellulose with sodium periodate is well-known to cleave the C2–C3 single bonds of the D-glucopyranose units (anhydroglucose unit, AGU), yielding so-called dialdehyde cellulose (DAC; Scheme 1). Several researchers have attempted to characterize the products of periodate-mediated cellulose oxidation. It has been determined that the concentration of free aldehyde groups in DAC or periodate-oxidized xylan is negligible because they are present in masked form as hemiacetals, (hemi)aldals, and aldehyde hydrates, which are subject to fast dynamic interconversion in aqueous solutions [13,14,15,16,17,18,19]. The periodate oxidation reaction can occur in any *cis*-vicinal diol structure in polysaccharides [20], so these masked aldehyde forms are also highly likely to be present in DAC.

We recently reported a fabrication method for ESNWs, which are composed of a highly periodate-oxidized cellulose and poly(vinyl alcohol) (PVA) [21]. This method involved four kinds of cellulosic materials from different sources. The methodology can be described as the direct electrospinning of DAC from aqueous solutions. The DAC preparations were soluble in water upon heating [22], and the heated aqueous solution turned slightly yellow. The DAC/PVA aqueous solutions tended to form hydrogels depending on the DAC concentration (ca. 5 wt%) [21]. Thus, it is evident that several improvements of the DAC preparation method and DAC/PVA dope properties are required for more reproducible and straightforward manufacturing of DAC-based ESNWs.

The present study was conducted to address these issues. Specifically, it explores (i) a general preparation method for DAC/PVA solutions, which are soluble in water at room temperature without the need for heating; (ii) a concept for avoiding any gelation of DAC/PVA aqueous solutions due to intermolecular hemiacetal formation [20]; (iii) a thorough material characterization of DAC/PVA ESNWs; and (iv) a post-spinning treatment for insolubilization of ESNWs with a diisocyanate [23], which can expand the scope of application for DAC thin fiber-based materials.

## 2. Materials and Methods

### 2.1. Materials

The cellulose was purchased from Sigma-Aldrich Japan (cellulose, fibrous, medium, from cotton, batch #098K0157; Tokyo). Sodium periodate, sodium hydrogen sulfite, Schiff’s reagent, *n*-propanal (98% GC), and other chemicals were obtained from Wako Pure Chemical Industry (Osaka, Japan). The PVA obtained from Wako had a degree of polymerization (DP) of 2000 and a degree of deacetylation of 78–82% mol.

### 2.2. Preparation of Dialdehyde Cellulose (DAC)

Cellulose powder (2.0 g) was dispersed in distilled/deionized water (DDW, 20 mL), and then the dispersion temperature was set to 55–60 °C. Sodium periodate (NaIO_4_, 5.28 g, 2.0 eq.mol/AGU) was added to the dispersion while stirring. The reaction was continued for 60–90 min, and then the reaction mixture was centrifuged (10,855× *g*, 25 min, room temperature). The supernatant was filtered and then dialyzed against DDW (MW cut-off ca. 6–8 kDa, Millipore dialysis tube) to remove inorganic salts. DAC was recovered as a white powder after lyophilization. The typical yield was 55–60 mass%. The prepared DAC samples were water-soluble at room temperature up to a DAC concentration of 35 wt%.

### 2.3. Viscosity Measurement

A Brookfield programmable rheometer (Middleboro, MA, USA), DV-III+, equipped with a small sample adaptor (spindle no. 14) was employed to record the apparent viscosity time-course of the reaction mixture as previously described. The reaction temperature was 55 ± 1 °C, and oxidation was performed in a small glass vessel with a water jacket, in which a heating medium was circulated. The average values at each time point were calculated from three independent reactions.

### 2.4. Quantification of Aldehyde Contents in DAC Preparations

To quantify the aldehyde contents of the DAC preparations, *n*-propanal (propionaldehyde) was used as a standard. The *n*-propanal stock solution (20 mM in DDW) was diluted to 0–2.0 mM in 2.0 vol% NaHSO_3_ (2.0 mL), and then 0.5 mL of Schiff’s reagent was added. After flush mixing for 15 min, standard curves were prepared based on the absorbance at 520 nm (Abs, 0 to ca 1.0) using a Hitachi UV2000 (Tokyo, Japan) spectrophotometer (cell length: 1.0 cm). DAC samples (ca 3.3–3.5 mg) were dissolved in DDW and diluted to ca. 140–150 µg/80 µL for dispersion in the quantification mixture, similar to the procedure used for the propionaldehyde standards. The aldehyde content of DAC in the present study is presented as a propanal equivalent in moles of aldehyde.

### 2.5. Electrospinning

A NANON-3 electrospinning apparatus (MECC, Inc., Fukuoka, Japan) was employed to prepare all ESNWs (Scheme 2). The tip-to-collector distance was set at 110–150 mm, and the applied voltage was fixed at 25 kV unless otherwise stated. Dope solutions typically containing 5 wt% DAC and 5 wt% PVA were loaded into a plastic syringe (5 mL), which was connected via a polytetrafluoroethylene tube (inner diameter: 1 mm) to an emitter with a Thermo 27 G needle (with a modified tip taper). A flat bed collector was covered with aluminum foil substrates for the recovery of ESNWs.

### 2.6. Scanning Electron Microscopy and Elemental Mapping

A Hitachi S3000 (Tokyo, Japan) scanning electron microscope (SEM) equipped with a Horiba EMAX energy dispersive X-ray (EDX) analysis probe (Kyoto, Japan) was employed for observation at low magnification (80×). Two DAC/PVA-ESNW samples were placed in parallel on carbon tape. One was spun from DDW, and the other was spun from 1 wt% NaHSO_3_. The EDX spectrum of each was acquired for 24 min at an energy range of 0.6–5.0 eV, in which EDX corresponds to the sodium and sulfur compounds. During observations at a magnification of 5000×, the fiber diameter values were randomly sampled from SEM images with up to 100 points (*n* = 100), and the statistical parameters including the mean, median, absolute difference between mean and median (*D*), and standard deviations were calculated. Significance between the mean values was calculated using the Student’s t-test.

### 2.7. Post-Spun Treatment of Electrospun Non-Woven Fabrics (ESNWs)

DAC/PVA ESNWs on the aluminum substrate were cut into pieces (1 cm × 2 cm) and dried in a vacuum desiccator. Hexamethylene diisocyanate (HMDI, 100 µL) was dissolved in absolute *n*-hexane (distilled from sodium, 400 µL), and then ca. 100–150 µL of the solution was added dropwise onto the surface of the ESNW, which was placed on a nylon mesh (200 mesh). After evaporation of the *n*-hexane, the ESNW was placed in a heating oven at 55–60 °C for 3 h. After removing the unreacted HMDI with *n*-hexane, the ESNW was air-dried and then further dried in a vacuum desiccator. The HMDI-treated ESNW was immersed in DDW for 1 h at 25 °C and air-dried again. Morphological changes before and after immersion in the DDW were observed via SEM in order to evaluate the network structure retention and average fiber diameters.

### 2.8. Vibrational Spectra

A Horiba F720 FTIR spectrometer (Kyoto, Japan) was employed to record the vibrational spectra of the ESNWs, and the samples were placed on a KRS5 prism (thallium bromide/iodide) in the attenuated total reflection (ATR) mode. The integrated accumulated spectra (8–16 scans) were processed by data acquisition software.

## 3. Results and Discussion

### 3.1. Water-Soluble DAC

Figure 1 presents the typical time-course of the apparent viscosity during oxidation of cellulose with sodium periodate. Initially, the cellulose was a dispersion (*t* = 0–10 min) and then it became a sticky sol as the apparent viscosity steeply increased to ca. 2.0 × 10^4^ mPa·s. This state was maintained from *t* = 30–40 min, and then the apparent viscosity gradually decreased until the mixture became a translucent solution at *t* = 60–80 min. Then, the apparent viscosity further decreased and reached the initial value, at which the reaction mixture was almost completely transparent.

The recovery yields of DAC with reaction times of 60–80 min and 100–140 min were 55–60 mass% and less than 35 mass%, respectively, suggesting that with longer reaction times, the oxidative degradation of cellulose becomes increasingly dominant, leading to the release of fragments with low molecular weight that are able to pass through the dialysis membrane (cut-off ca. 6500 g mol^−1^).

The aldehyde content of DAC was, on average, 11.1 ± 0.2 mmol/g (*n* = 4), which corresponds to 89 ± 1 mol% of the theoretical value (12.5 mmol/g). The prepared DAC was soluble in water at room temperature at concentrations up to 35 wt%, which is favorable for subsequent electrospinning. For DAC concentrations above 10 wt%, the aqueous solution formed a hydrogel after a few hours, which is problematic for electrospinning operations.

### 3.2. Preparation of Dialdehyde Cellulose/Poly(Vinyl Alcohol) (DAC/PVA) Dope for Electrospinning

At the lower concentration of 5 wt%, the DAC solutions retained in a sol state, but were not viscous enough to develop into thin fibers via the conventional electrospinning procedure. Therefore, PVA was added to the dope at a concentration of 5 wt%, which successfully created submicron-scale thin fibers, as described in a previous study [21]. The DAC/PVA 5 wt%/5 wt% mixture, however, slowly transformed into a hydrogel upon storage for over 24 h. Hence, DAC/PVA aqueous solutions must be immediately electrospun after dope preparation. This limitation will be addressed as follows.

Sodium hydrogen sulfite (NaHSO_3_) is a well-known reagent that undergoes reactions with aldehyde functionalities to yield bisulfite addition products (Scheme 3). The gelation of DAC is triggered by inter-chain crosslinking through hemiacetal formation [20] between cellulose hydroxyls and aldehyde groups. The presence of NaHSO_3_ in the dope suppressed hemiacetal formation, resulting in inhibition of dope gelation. Table 1 summarizes the characteristics of solutions with NaHSO_3_ concentrations from 0–10 wt% in the dope (DAC/PVA = 5 wt%/5 wt%). Higher NaHSO_3_ concentrations decreased the pH value of the solution, which significantly promoted hemiacetal formation.

Table 1 shows the dope 24 h after preparation without the addition of NaHSO_3_. The dope evidently formed a hydrogel at a NaHSO_3_ concentration of 0 wt%. When the pH values were below 3.0 (NaHSO_3_ ≥ 2.5 wt%), white turbid aggregates were generated. Thus, concentrations of NaHSO_3_ above 2.5 wt% are not suitable for electrospinning operations. At a NaHSO_3_ concentration of 1 wt%, the dope retained a transparent sol state, and this solution contained 17 mol% aldehyde bisulfite adducts in the DAC solution. Based on these observations, a DAC/PVA 5 wt%/5 wt% dope containing 1 wt% NaHSO_3_ was used for subsequent ESNW fabrication.

### 3.3. Characterization of DAC/PVA ESNWs

The morphologies of DAC/PVA fibrous matrices spun from an aqueous solution without NaHSO_3_ (DDW) (Figure 2a) and with 1 wt% NaHSO_3_ (Figure 2b) were composed of thin fibers with average diameters (*ϕ* ± standard deviation) of 347 ± 47 nm and 434 ± 158 nm, respectively. The difference in the *ϕ* values were statistically significant based on the probability value (*p* < 0.001). This indicated that the presence of 1 wt% NaHSO_3_ resulted in thicker fibers upon electrospinning, which might be caused by the partially broken charge separation [24] due to the presence of electrolytes, bisulfite adducts, and sodium cations. The absolute differences between the average and median (*D*) were 8.2 nm and 38 nm in the absence and presence of 1 wt% NaHSO_3_, respectively. Taken together with the standard deviations of *ϕ*, this result indicates that DAC/PVA ESNW spun from 0 wt% exhibited more homogeneous fiber diameters.

To support the presence of aldehyde-bisulfite adducts in the DAC/PVA ESNW spun from a 1 wt% NaHSO_3_ solution, SEM-EDX analysis was performed. Figure 3a represents an SEM image (80× magnification) of the two DAC/PVA specimens spun from 0 and 1 wt% NaHSO_3_ solutions. The energy dispersion spectra were collected for the areas specified in the boxes in Figure 3a. The elemental mapping images (Figure 3b) indicated that the DAC/PVA ESNW spun from 1 wt% NaHSO_3_ contained sulfur, while the ESNW spun from 0 wt% NaHSO_3_ did not. The signal of sodium was weak for the ESNW spun from 0 wt% NaHSO_3_ (Figure 3c), likely due to the trace amounts of Na remaining from the oxidation reaction with NaIO_4_. In contrast, the ESNW spun from 1 wt% NaHSO_3_ exhibited pronounced Na signal intensity. Figure 3d shows the EDX spectra of both ESNWs, in which a strong signal of the aluminum foil substrate was observed as well as the coincident signals from S and Na. Subtraction of spectrum 2 (gray line in Figure 3d; spun from 0 wt% NaHSO_3_) from spectrum 1 (red line in Figure 3d; spun from 1 wt% NaHSO_3_) left significant signals for the S and Na (blue line in Figure 3d). This indicates that the DAC/PVA ESNW spun from 1 wt% NaHSO_3_ was composed of DAC molecules in which the aldehyde groups were partly modified to become bisulfite adducts.

### 3.4. Post-Spun Treatment for Insolubilization of DAC/PVA ESNWs

The DAC/PVA ESNWs spun from 0 wt% and 1 wt% NaHSO_3_ solutions cannot retain their network structures in water because both DAC and PVA are water-soluble materials. Therefore, a chemical treatment for intermolecular cross-linking was applied to the as-spun ESNWs. HMDI is chemically reactive toward the hydroxyl groups in DAC and PVA and establishes urethane-type crosslinks. It does not directly react with aldehyde groups. Figure 4 shows changes in the infrared spectra of the DAC/PVA ESNW before and after the HMDI treatment.

A series of chemical conversions from the initial cellulose powder to the final DAC/PVA ESNWs can be confirmed by vibrational spectral changes. The DAC preparation (green line in Figure 4) exhibited stretching vibration of the aldehyde C=O at 1740 cm^−1^, while the initial cellulose powder (black line in Figure 4) did not, corresponding to the expected progress in oxidation. The signal is not extremely pronounced since most aldehydes in DAC are present in a masked form, as above-mentioned. After preparation of the DAC/PVA ESNW spun from 0 wt% NaHSO_3_, the vibrational spectrum of the ESNW was similar to that of DAC (data not shown).

When the DAC/PVA ESNW spun from 0 wt% NaHSO_3_ was treated with HMDI for insolubilization, the OH groups of the DAC and PVA molecules reacted with the isocyanate moieties, –N=C=O, to form urethane-type crosslinks, –NH–C(=O)–O–. The development of coupling vibrational absorption of *ν*_C = O_ + *δ*_N-H_ at 1650 cm^−1^ (blue line in Figure 4) supports the occurrence of this chemical conversion. The HMDI-treated ESNW spun from 1 wt% NaHSO_3_ features the same vibrational absorption (red line in Figure 4). Excess HMDI leads to some unreacted NCO groups, as evidenced by the stretching vibration of 2280 cm^−1^ in both spectra of the ESNW spun from 0 wt% and 1 wt% NaHSO_3_ solutions.

Figure 5 presents the morphologies of the DAC/PVA ESNW spun from 1 wt% NaHSO_3_ after the HMDI treatment (Figure 5a, boxplot 2) and after immersion in the DDW (Figure 5b, boxplot 3). The thin fiber network structure of the DAC/PVA-ESNW was retained after HMDI treatment, and the average fiber diameter (*ϕ*) values (*n* = 100) before and after the treatment (434 ± 159 nm and *ϕ* = 442 ± 125 nm, respectively) were unchanged (within the measurement error; boxplots 1 and 2 in Figure 5; *p* = 0.72). The *D* value of the cross-linked ESNW (6.5 nm; boxplot 2 in Figure 5) indicated a narrow distribution. After immersion in the DDW, as seen in Figure 5b, the fiber network partly deformed, but the fibers remained attached at their crossing points, resulting in the formation of tiny pores of ca. 6–17 µm across the material with curved peripherals. The *ϕ* value (*n* = 100) and degree of diameter distribution after immersion in the DDW statistically increased to 577 ± 204 nm (boxplot 3 in Figure 5; *D* = 47 nm, *p* < 0.001). This *ϕ* value was significantly higher than that of the HMDI-treated ESNW spun from 0 wt% NaHSO_3_ after DDW immersion (512 ± 136 nm; *n* = 100, *D* = 38 nm; boxplot 4 in Figure 5; *p* = 0.0084).

## 4. Conclusions

The DAC/PVA dope prepared with 1 wt% NaHSO_3_ was successfully used to create ESNWs with single-fiber diameters in the sub-micron range. In this process, PVA molecules probably not only act as a thickening agent that can be used to achieve appropriate viscosity for the electrospinning procedure, but also as a nonionic surfactant that forms insulating layers on the surfaces of the tiny droplets generated when a voltage is applied [24]. Even in the presence of bisulfite adducts and Na^+^ as ionic electrolytes, this insulating layer plays a key role in generating the drawing force to spin thin fibers according to the charge separation principle.

HMDI-mediated cross-linking of the DAC/PVA thin fibers makes the ESNWs water-insoluble while retaining the thin fiber network structures as well as preserving masked aldehyde groups as reactive sites for further modification, as seen in the vibrational spectra. The 83 mol% of the masked aldehydes, estimated as 9.21 mmol/g DAC, remained available for follow-up chemistry including between aldehydes and primary amino groups on the biologically active peptides. This result will be addressed in our upcoming studies and reports.

A promising application of the DAC/ESNW is as a scaffolding material for cell culturing in tissue engineering. This conclusion is connected to our previous reports on phosphorylated peptide synthesis [25,26,27]. These reports addressed, for example, the use of *O*-phospho-L-serine-conjugated DAC/PVA ESNW-based scaffolds for hard tissue regeneration as a way to bypass the time-consuming preparation steps of phosphopeptide–cellulose conjugate synthesis [28,29,30,31], and the possibility that aldehyde functionalities in the DAC/PVA ESNW matrices could promote ligand immobilization by a simple procedure that involves dipping ESNWs into a dyeing pad [32]. Based on the multitude of follow-up reactions of the contained masked aldehyde groups, DAC/PVA ESNWs are multipurpose materials for conjugating/immobilizing various kinds of biologically active substances [33] or metal oxide nanoparticles [34,35]. This result could inspire tailor-made designs of functionalized ESNWs for novel nano-scaled materials.

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
