# Peer review of "A General Protocol for Electrospun Non-Woven Fabrics of Dialdehyde Cellulose and Poly(Vinyl Alcohol)"

_nanomaterials, 2020, doi:10.3390/nano10040671_

Round 1

Reviewer 1 Report

This manuscript was well written. But several points are not clear yet. Firstly, it is not clear why author used DAC even with multiple and complicated reaction steps. In contrast, well electrospun PVA nanofiber membranes can be easily cross-linked via a HMDI treatment too. Also general cross-linker, such as glutaraldehyde, etc for the PVA can be also utilized in the straightforward way to produce the cross-linked PVA nanofibers. Secondly, what is the optimized cross-linking density for the DAC/PVA ESNW, which might be important for the application scopes of the DAC thin fiber-based materials. Moreover, it could be also useful to provide the mechanical properties and degree of swelling of the DAC/PVA ESNW before and after the swelling in water.

Author Response

ANSWER: The authors are thankful for the meaningful comments of reviewer #1. We would like to point out that our focus on DAC, its chemistry and utilization. The present work was done to establish a better DAC preparation method, which is also adaptable for the subsequent electrospinning process. As you may know, cellulose is a water-insoluble polymer, while DAC is water-soluble, so that usually cellulose preparations cannot be combined with PVA in the forms of aqueous solutions to be fabricated into nanofibers via electrospinning process.

As to the cross-linking reagents, glutaraldehyde is commercially available typically as "20-25 % aqueous solutions". Using this reagent, the fine fiber networks in the DAC/PVA ESNW matrices will not retain their fibrous shapes upon immersion in the aqueous glutaraldehyde solutions for cross-linking treatments. On the other hands, the DAC/PVA is insoluble in hydrophobic organic solvents, i.e., n-hexane, so that HMDI is adaptable for the purpose of cross-linking (insolubilization) of the DAC/PVA ESNWs.

The optimized cross-linking density and the corresponding degree of swelling, as pointed out, are performance parameters/indicators of the DAC/PVA ESNWs, and those parameters, which are related to the mechanical properties, will be reported in a separated article, after we will have established an accurate/appropriate methodology for crosslinker density quantification.

Thank you again for the helpful comments.

Reviewer 2 Report

This is an interesting article on the use of electrospinning. In this work, authors have reported on the preparation of the DAC/PVA dope in the presence of 1 wt% NaHSO3. The article is well-written. Following minor corrections are needed:

-Revise the article for language corrections

-There should be  a schematic for electrospinning

-Have the authors studied the effect of degree of polymerisation on electrospinning

-Relevant articles on electrospinning may be cited such as Nanomaterials 202010(3), 523; Nanomaterials 20199(3), 404; 

Nanomaterials 202010(3), 517

Author Response

ANSWER: The authors thank for the suggestions of reviewer #2, which all helped to improve the original manuscript. All of comments/suggestions have been incorporated in the revised manuscript.

-Revise the article for language corrections

ANSWER: The whole manuscript has been checked by an authorized Editing service and appropriate language corrections have been made throughout.

-There should be a schematic for electrospinning

ANSWER: A new Scheme 3 has been added in the revised manuscript to indicate the electrospinning apparatus and setup (NANON3, MECC, Fukuoka, Japan).

-Have the authors studied the effect of degree of polymerisation on electrospinning ?

ANSWER: The effect of polymerization on electrospinning process is not our focus in the present study, so that both PVA and cellulose preparations were used as received, while the material specifications are described in the section 2.1. The degree of polymerization of DAC and its measurement have been addressed in detail in our previous publications, so that a repetition did not seem necessary.

-Relevant articles on electrospinning may be cited such as Nanomaterials 2020, 10(3), 523; Nanomaterials 2019, 9(3), 404; Nanomaterials 2020, 10(3), 517

ANSWER: The author thanks for these suggestions. We agree that these references might be relevant to our study. Three references have been cited now in the revised manuscript as reference numbers [33], [34], and [35].

Thank you again for the helpful comments from the reviewer #2.